# Hepatitis C care cascade among patients with and without tuberculosis: Nationwide observational cohort study in the country of Georgia, 2015–2020

Davit Baliashvili[1]*, Henry M. Blumberg[1,2,3], Neel R. Gandhi[1,2,3], Francisco Averhoff[4], David Benkeser[5], Shaun Shadaker[6], Lia Gvinjilia[7], Aleksandre Turdziladze[8], Nestani Tukvadze[9], Mamuka Chincharauli[9], Maia Butsashvili[10], Lali Sharvadze[11,12], Tengiz Tsertsvadze[13], Jaba Zarkua[14], Russell R. Kempker[3]

1 Department of Epidemiology, Emory University Rollins School of Public Health, Atlanta, Georgia, United States of America, 2 Department of Global Health, Emory University Rollins School of Public Health, Atlanta, Georgia, United States of America, 3 Division of Infectious Diseases, Department of Medicine, Emory University School of Medicine, Atlanta, Georgia, United States of America, 4 Department of Family and Preventive Medicine, Emory University School of Medicine, Atlanta, Georgia, United States of America, 5 Department of Biostatistics and Bioinformatics, Emory University Rollins School of Public Health, Atlanta, Georgia, United States of America, 6 Division of Viral Hepatitis, Centers for Disease Control and Prevention, Atlanta, Georgia, United States of America, 7 Eastern Europe and Central Asia Regional Office, Centers for Disease Control and Prevention, Tbilisi, Georgia, 8 National Center for Disease Control and Public Health, Tbilisi, Georgia, 9 National Center for Tuberculosis and Lung Diseases, Tbilisi, Georgia, 10 Health Research Union and Clinic NeoLab, Tbilisi, Georgia, 11 Clinic "Hepa", Tbilisi, Georgia, 12 The University of Georgia, Tbilisi, Georgia, 13 Infectious Diseases, AIDS and Clinical Immunology Research Center, Tbilisi, Georgia, 14 Clinic "Mrcheveli", Tbilisi, Georgia

* dato.baliashvili@gmail.com

## Abstract

### Background

The Eastern European country of Georgia initiated a nationwide hepatitis C virus (HCV) elimination program in 2015 to address a high burden of infection. Screening for HCV infection through antibody testing was integrated into multiple existing programs, including the National Tuberculosis Program (NTP). We sought to compare the hepatitis C care cascade among patients with and without tuberculosis (TB) diagnosis in Georgia between 2015 and 2019 and to identify factors associated with loss to follow-up (LTFU) in hepatitis C care among patients with TB.

### Methods and findings

Using national ID numbers, we merged databases of the HCV elimination program, NTP, and national death registry from January 1, 2015 to September 30, 2020. The study population included 11,985 adults (aged ≥18 years) diagnosed with active TB from January 1, 2015 through December 31, 2019, and 1,849,820 adults tested for HCV antibodies between January 1, 2015 and September 30, 2020, who were not diagnosed with TB during that time. We estimated the proportion of patients with and without TB who were LTFU at each step of the HCV care cascade and explored temporal changes. Among 11,985 patients with

**Data Availability Statement:** The data used in this analysis are not owned by the authors and therefore cannot be shared publicly. The requests for the data sharing should be sent to the National Center for Disease Control and Public Health of Georgia via email: pr.ncdc@ncdc.ge, and to the

National Center for Tuberculosis and Lung Disease via email: tbcenter@tbgeo.ge. We confirm that others researchers who meet the criteria for access to confidential data will be able to access these data in the same manner as the authors. The authors did not have any special access privileges that others would not have.

**Funding:** This work was supported in part from grants from the National Institutes of Health (NIH) Fogarty International Center (D43TW007124; "Emory-Georgia TB Research Training Program") to National Institutue/National Institutue of Allergy and Infectious Diseases (K24AI114444; the Emory/Georgia TB Research Advancement Center, P30AI168386; the Emory University Center for AIDS Research, P30AI050409; and TB Research Unit-ASTRa, U19AI111211) to NRG. The funders had no role in study design, data collection and analysis, decision to publish, or preparation of the manuscript.

**Competing interests:** The authors have declared that no competing interests exist.

**Abbreviations:** aRR, adjusted risk ratio; CI, confidence intervals; cRR, crude risk ratio; DAA, directly-acting antiviral; DAG, directed acyclic graph; DS, drug-susceptible; HCV, hepatitis C virus; HIV, human immunodeficiency virus; HR, hazards ratio; IDP, internally displaced person; IQR, interquartile range; LTFU, loss to follow-up; MDR, multidrug-resistant; NTP, National Tuberculosis Program; PCR, polymerase chain reaction; RNA, ribonucleic acid; SVR, sustained virologic response; TB, tuberculosis; Tx, treatment.

active TB, 9,065 (76%) patients without prior hepatitis C treatment were tested for HCV antibodies, of which 1,665 (18%) had a positive result; LTFU from hepatitis C care was common, with 316 of 1,557 (20%) patients with a positive antibody test not undergoing viremia testing and 443 of 1,025 (43%) patients with viremia not starting treatment for hepatitis C. Overall, among persons with confirmed viremic HCV infection, due to LTFU at various stages of the care cascade only 28% of patients with TB had a documented cure from HCV infection, compared to 55% among patients without TB. LTFU after positive antibody testing substantially decreased in the last 3 years, from 32% among patients diagnosed with TB in 2017 to 12% among those diagnosed in 2019. After a positive HCV antibody test, patients without TB had viremia testing sooner than patients with TB (hazards ratio [HR] = 1.46, 95% confidence intervals [CI] [1.39, 1.54], $p < 0.001$). After a positive viremia test, patients without TB started hepatitis C treatment sooner than patients with TB (HR = 2.05, 95% CI [1.87, 2.25], $p < 0.001$). In the risk factor analysis adjusted for age, sex, and case definition (new versus previously treated), multidrug-resistant (MDR) TB was associated with an increased risk of LTFU after a positive HCV antibody test (adjusted risk ratio [aRR] = 1.41, 95% CI [1.12, 1.76], $p = 0.003$). The main limitation of this study was that due to the reliance on existing electronic databases, we were unable to account for the impact of all confounding factors in some of the analyses.

## Conclusions

LTFU from hepatitis C care after a positive antibody or viremia test was high and more common among patients with TB than in those without TB. Better integration of TB and hepatitis C care systems can potentially reduce LTFU and improve patient outcomes both in Georgia and other countries that are initiating or scaling up their nationwide hepatitis C control efforts and striving to provide personalized TB treatment.

## Author summary

### Why was this study done?

- There is ample evidence that hepatitis C prevalence is disproportionally high among patients with tuberculosis (TB).

- Highly effective new treatment options for hepatitis C allowed many countries, including Georgia, to implement large-scale hepatitis C programs.

- It has not been well characterized how often patients with current or past TB are offered and provided with hepatitis C testing and treatment services.

### What did the researchers do and find?

- We conducted an observational cohort study comparing the hepatitis C care cascade among patients with and without TB to explore if patients with tuberculosis receive hepatitis C treatment completely and timely.

- The proportion of patients with TB tested for hepatitis C virus (HCV) antibodies increased per year. Among patients diagnosed with TB in 2015, 60% were tested for HCV antibodies sometime during the study period. This proportion reached 90% among patients diagnosed with TB in 2019

- Loss to follow-up (LTFU) from hepatitis C care was more common among patients with TB, with 20% of patients with a positive antibody test not undergoing viremia testing and 43% of patients with viremia not starting treatment for hepatitis C. For comparison, the respective numbers among patients without TB were 14% and 19%.

**What do these findings mean**

- Our findings highlight the importance of improving integration and linkage to hepatitis C diagnostic and treatment services among patients with TB.

- Existing large-scale public health programs for both TB and hepatitis C in Georgia and other countries with nationwide programs create a unique opportunity for integrated care of these 2 infectious diseases, which could potentially reduce LTFU and improve overall health outcomes.

## Introduction

Both tuberculosis (TB) and hepatitis C virus (HCV) infection cause substantial morbidity and mortality worldwide. In 2020, there were an estimated 10 million new cases of active TB globally and 1.5 million deaths due to TB [1]. There are an estimated 59 million people living with chronic HCV infection, 1.5 million new cases occur each year, and 290,000 people died due to hepatitis C globally in 2019 [2]. TB and HCV infection are often concentrated in the same high-risk population subgroups, such as homeless or incarcerated individuals and people who inject drugs [3–5]. However, it has not been well characterized how often patients with current or past TB are offered and provided with hepatitis C testing and treatment services.

Hepatitis C diagnosis, treatment, and assessment of cure consist of multiple consecutive steps, collectively known as the hepatitis C care cascade [6]. The cascade starts with hepatitis C antibody testing and ends with the determination of cure based on sustained virologic response (SVR) at 12 weeks after treatment completion [7–16]. Available data has found that loss to follow-up (LTFU) at different stages of the hepatitis C care cascade is common [17]. In 2019, globally, only 26% of people living with hepatitis C knew their infection status, and only 16% were treated [2].

The hepatitis C care cascade among patients with both TB and hepatitis C has not been previously studied. However, given the increased rate of adverse events including hepatoxicity among patients with TB and HCV coinfection, screening and potentially early treatment should be a high priority in this group [18,19]. Historically, patients receiving treatment for TB have not been recommended to receive hepatitis C treatment until they finish TB treatment due to immunosuppressive effect of interferon or due to drug–drug interactions of more recently introduced directly-acting antivirals (DAAs) [9,20]. However, the combined effect of TB drugs and HCV infection can cause long-term impairment of liver function, and timely

management of hepatitis C in patients with TB as soon as they become eligible should be an important research priority. Since patients with TB are already receiving care, there is a key opportunity to engage them in hepatitis C care without interruption, reducing the likelihood of LTFU.

The Eastern European country of Georgia (population 3.7 million) is designated by the WHO as a high-priority country for TB control in European region, with an estimated incidence of 70 TB cases per 100,000 population in 2020 and a high percentage of multidrug-resistant (MDR) TB [1,21]. Chronic hepatitis C is also highly prevalent in Georgia, affecting an estimated 5.4% of the general adult population (approximately 150,000 individuals) based on a 2015 serosurvey—the highest prevalence among countries of Eastern Europe and Central Asia [22,23]. Historically, there has been a substantial overlap in the population affected by these 2 infectious diseases in Georgia; previous studies reported that >20% of newly diagnosed active TB cases had HCV antibodies, indicative of current or past infection [18,24].

To address the high burden of HCV infection, Georgia initiated the first nationwide hepatitis C elimination program in 2015, including free hepatitis C testing and treatment with new DAAs for all citizens [25–27]. The cost of confirmatory viremia testing was initially fully covered by the government only for persons with low income but became free for everyone beginning in March 2018 [28]. Free hepatitis C screening through antibody testing was gradually integrated into multiple existing programs and settings, including the National TB Program (NTP), which provides free diagnostic and treatment services for TB countrywide for residents of Georgia [26,28–31]. Before 2018, HCV antibody testing was performed routinely among patients with MDR TB and sporadically (by indication) among patients with drug-susceptible (DS) TB. According to the TB treatment guideline in Georgia adopted in July 2018, all patients newly diagnosed with TB should be routinely tested for HCV antibodies. If an antibody test is positive, blood samples are sent to the National Center for Disease Control and Public Health for HCV viremia testing. However, currently there is no formal referral system in place linking viremia–positive patients with TB to hepatitis C care.

The objectives of this study were to (1) compare the hepatitis C care cascade among patients with and without TB diagnosis in Georgia between 2015 and 2019. We hypothesized that the proportion of HCV seropositive individuals who undergo HCV confirmatory testing, initiate and complete HCV treatment was lower among patients with TB compared to patients without TB; and (2) identify factors associated with LTFU in hepatitis C care among patients with TB. Findings from this analysis will be a valuable contribution to the Georgian Hepatitis C Elimination Program and other similar settings by providing data for future targeted interventions to improve linkage to and retention in hepatitis C care among patients with TB.

## Methods

### Study design and setting

We conducted an observational cohort study and analyzed the hepatitis C care cascade among patients diagnosed with active TB disease and compared it to the care cascade among people with HCV without active TB disease. TB-related information was obtained from the NTP of Georgia. Hepatitis C screening, diagnostic, and treatment information was obtained from the Georgian Hepatitis C Elimination Program.

### Study population

The study population consisted of 2 groups: (1) Adults (aged ≥18 years) diagnosed with active TB through the Georgian NTP from January 1, 2015 through December 31, 2019; and (2) adults tested for HCV antibodies in Georgia between January 1, 2015 and September 30, 2020,

who were not diagnosed with TB during that time. We did not include patients diagnosed with TB in 2020 since they would have very limited follow-up time for HCV-related procedures. For both groups, hepatitis C testing and treatment information was obtained through September 30, 2020.

### Hepatitis C care cascade steps and definitions

We assessed the following steps in the hepatitis C care cascade (Fig 1): (1) Screening for HCV antibodies using either rapid or laboratory-based tests; (2) viremia testing via ribonucleic acid (RNA) or core-antigen testing to confirm viremic HCV infection among those with HCV antibody positive result; (3) hepatitis C treatment initiation among those with positive viremia test; (4) hepatitis C treatment completion; and (5) SVR testing from 12 to 24 weeks after treatment completion.

Two separate care cascade analyses were performed among persons with HCV infection with and without TB disease. The hepatitis C care cascade among patients with TB included the following groups: (1) Persons tested for HCV antibodies at the time of or any time after their first TB diagnosis; and (2) persons tested for HCV antibodies before their TB diagnosis who did not initiate treatment for hepatitis C before their TB diagnosis, i.e., people who were eligible for hepatitis C care by the time they were enrolled in TB treatment. Hepatitis C care cascade among patients without TB included the following groups: (1) Persons tested for HCV antibodies who were not diagnosed with TB during 2015 to 2019; and (2) persons diagnosed with and treated for chronic HCV infection before they were first diagnosed with TB (thus, did not have TB at the time of their HCV diagnosis and treatment).

Patients were defined as LTFU after a positive antibody test if they had a positive HCV antibody test but did not undergo HCV viremia testing within the HCV elimination program and were still alive at the time of data extraction (September 30, 2020). Patients were defined as LTFU after an HCV viremia test if they had a positive result on viremia testing (i.e., positive RNA or core antigen test) and did not start treatment for HCV infection within the HCV elimination program. Due to a lack of complete drug-susceptibility testing data, patients with TB were defined as having DR TB if they received TB treatment with second-line drugs which are used to treat MDR-TB.

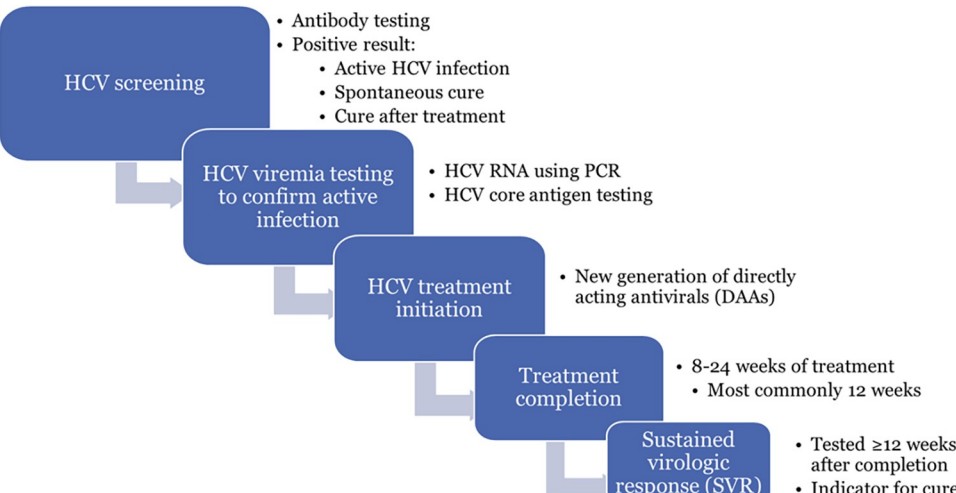

**Fig 1. Steps of hepatitis C care cascade in Georgian Hepatitis C Elimination Program.** DAA, directly-acting antiviral; HCV, hepatitis C virus; PCR, polymerase chain reaction; RNA, ribonucleic acid; SVR, sustained virologic response.

## Sources of data

Four data sources were utilized. We obtained all TB-related data from the Georgian NTP surveillance database. This database contains diagnostic and treatment-related information on every patient enrolled in the NTP, including penitentiary system. HCV screening information was obtained from the national HCV screening registry—a real-time, nationwide web-based system. The screening registry collects data from all sites providing HCV screening throughout the country, including TB facilities. HCV viremia testing and treatment information were obtained from the Hepatitis C Elimination Program clinical database, called "Elimination C" (ElimC) [32]. ElimC collects information related to viremia testing, other diagnostic procedures, and antiviral treatment from all service providers, except the medical facilities in penitentiary system. Patients included in any of these databases were cross-checked in the National Death Registry, and death dates were obtained for deceased patients. A unique national ID number was used as a linking variable between all data sources.

## Statistical analysis

We conducted descriptive statistics of patients with and without TB to describe available demographic and clinical characteristics (Tables 1–3). Due to a limited number of variables in HCV screening registry, we could not conduct extensive comparison of the 2 groups. In the care cascade analysis, we first calculated the proportions of people reaching each step of the hepatitis C care cascade, as described above. Next, we used survival analysis methods to explore if the time from a positive HCV antibody test to viremia testing and time from a positive viremia test to hepatitis C treatment initiation differed between patients with and without TB. For patients with TB who had positive viremia test during their TB treatment, follow-up time started at the end of their TB treatment. Patients were censored at the end of the follow-up period (September 30, 2020). We used subdistribution hazards model to calculate hazards ratios (HR) and assess the differences between groups numerically. Death was treated as a competing risk. We also created cumulative incidence curves to examine the differences graphically, and Gray's test for equality of cumulative incidence functions was used to test for differences [33,34]. Among those who underwent viremia testing, median time from antibody testing to viremia testing was compared using Wilcoxon rank-sum test.

An additional analysis was performed among patients with TB to identify demographic or TB-related factors associated with LTFU in hepatitis C care after a positive antibody or viremia test. The analysis of factors associated with LTFU after a positive antibody test included all patients with TB (both DS and DR) who had a positive HCV antibody test result. The analysis of factors associated with LTFU after a positive viremia test was restricted to patients with DS TB who completed their TB treatment. Patients still on DS TB treatment were excluded because they are ineligible for hepatitis C treatment until after TB treatment is completed. Patients with DR TB were excluded due to heterogeneous TB treatment duration and lack of a standardized approach regarding the timing of hepatitis C treatment initiation. Log-binomial regression was used to calculate adjusted risk ratios (aRR) and 95% confidence intervals (CI). Covariate selection was conducted a priori based on directed acyclic graph (DAG) theory (S1–S4 Figs) [35]. Different sets of covariates were used for each of the patient characteristics. Significance level of 0.05 was used in all analyses that involved significance testing. To address the missing data in some of the covariates, we conducted multiple imputation with fully conditional specification using the same covariates as in the substantive analysis (20 imputations) [36]. All analyses were performed using SAS version 9.4.

**Table 1. Demographic and clinical characteristics of patients diagnosed with TB, Georgia, January 1 2015–December 31, 2019.**

| Variable | N | % |
|---|---|---|
| **Demographic characteristics** | | |
| **Sex** | | |
| Male | 8,368 | 69.8 |
| Female | 3,617 | 30.2 |
| **Age** | | |
| Median (IQR) | 42 (25) | |
| **Region** | | |
| Tbilisi | 5,589 | 46.6 |
| Kakheti | 474 | 4.0 |
| Shida Kartli | 500 | 4.2 |
| Kvemo Kartli | 883 | 7.4 |
| Mtskheta-Mtianeti | 148 | 1.2 |
| Imereti | 1,286 | 10.7 |
| Guria | 192 | 1.6 |
| Achara | 1,305 | 10.9 |
| Samegrelo | 1,228 | 10.3 |
| Racha | 21 | 0.2 |
| Samtskhe | 168 | 1.4 |
| Penitentiary system | 191 | 1.6 |
| **Employment status** | | |
| Employed | 1,677 | 14.0 |
| Unemployed | 9,815 | 81.9 |
| Military | 25 | 0.2 |
| Missing | 468 | 3.9 |
| **IDP** | | |
| No | 10766 | 89.8 |
| Yes | 700 | 5.8 |
| Missing | 519 | 4.3 |
| **History of imprisonment** | | |
| No | 10743 | 89.6 |
| Yes | 799 | 6.7 |
| Missing | 443 | 3.7 |
| **TB diagnosis and treatment** | | |
| **Year of TB diagnosis** | | |
| 2015 | 2,766 | 23.1 |
| 2016 | 2,740 | 22.9 |
| 2017 | 2,405 | 20.1 |
| 2018 | 2,086 | 17.4 |
| 2019 | 1,988 | 16.6 |
| **Number of TB treatment episodes** | | |
| 1 | 10,844 | 90.5 |
| 2 | 951 | 7.9 |
| 3 | 148 | 1.2 |
| 4 | 28 | 0.2 |
| 5 | 7 | 0.1 |
| 6 | 6 | 0.1 |

*(Continued)*

**Table 1.** (Continued)

| Variable | N | % |
|---|---|---|
| 7 | 0 | 0.0 |
| 8 | 1 | 0.0 |
| **TB case definition** | | |
| New | 9,615 | 80.2 |
| Other/Unknown | 2,370 | 19.8 |
| **Past TB treatment outcome** | | |
| Successful | 1,418 | 69.1 |
| Unsuccessful/Unknown | 634 | 30.9 |
| **On second-line treatment** | | |
| Yes | 1,281 | 10.7 |
| No | 10,704 | 89.3 |

IDP, internally displaced persons; IQR, interquartile range; TB, tuberculosis.

## Ethics

The study was approved by the institutional review boards (IRBs) at the National Center for Disease Control and Public Health (Tbilisi, Georgia), the National Center for Tuberculosis and Lung Disease (Tbilisi, Georgia), and Emory University (Atlanta, Georgia, United States of America). CDC's Human Subjects Research Office determined CDC's role to be provision of technical assistance that did not constitute engagement in the research. This study is reported as per the Strengthening the Reporting of Observational Studies in Epidemiology (STROBE) guideline (S1 Checklist).

**Table 2. Baseline demographic characteristics of persons tested for HCV antibodies, Georgia, January 1, 2015– September 30, 2020.**

| Variable | N | % |
|---|---|---|
| **Sex** | | |
| Male | 836,736 | 45.23 |
| Female | 1,013,084 | 54.77 |
| **Age** | | |
| Median (IQR) | 46 (31–62) | |
| **Region** | | |
| Adjara | 198,705 | 10.74 |
| Guria | 50,561 | 2.73 |
| Imereti | 228,590 | 12.36 |
| Kakheti | 105,308 | 5.69 |
| Kvemo Kartli | 80,975 | 4.38 |
| Mtskheta-Mtianeti | 23,914 | 1.29 |
| Racha-Lechkhumi-Kvemo Svaneti | 11,535 | 0.62 |
| Samegrelo-Zemo Svaneti | 170,557 | 9.22 |
| Samtskhe-Javakheti | 48,130 | 2.6 |
| Shida Kartli | 71,954 | 3.89 |
| Tbilisi | 825,900 | 44.65 |
| Missing | 33,691 | 1.82 |

HCV, hepatitis C virus; IQR, interquartile range.

**Table 3. Baseline demographic characteristics of persons tested positive for HCV antibodies, Georgia, January 1, 2015–September 30, 2020.**

| Variable | N | % |
|---|---|---|
| **Sex** | | |
| Male | 96,050 | 73.41 |
| Female | 34,787 | 26.59 |
| **Age** | | |
| Median (IQR) | 47 (39–57) | |
| **Region** | | |
| Adjara | 7,655 | 5.85 |
| Guria | 1,880 | 1.44 |
| Imereti | 13,022 | 9.95 |
| Kakheti | 2,536 | 1.94 |
| Kvemo Kartli | 2,902 | 2.22 |
| Mtskheta-Mtianeti | 848 | 0.65 |
| Racha-Lechkhumi-Kvemo Svaneti | 413 | 0.32 |
| Samegrelo-Zemo Svaneti | 10,646 | 8.14 |
| Samtskhe-Javakheti | 686 | 0.52 |
| Shida Kartli | 3,259 | 2.49 |
| Tbilisi | 82,621 | 63.15 |
| Missing | 4,369 | 3.34 |

HCV, hepatitis C virus; IQR, interquartile range.

## Inclusivity in global research

Additional information regarding the ethical, cultural, and scientific considerations specific to inclusivity in global research is included in the Supporting information (S2 Checklist).

## Results

### Participants

A total of 14,993 records were obtained from the NTP database; 913 records (6.1%) were excluded due to a missing national ID number (Fig 2). The remaining 14,080 records corresponded to 12,767 individual patients, of whom 11,985 (94%) were aged ≥18 years and included in the analysis. Among those included (*n* = 11,985), 70% were males, 82% were unemployed at the time of TB diagnosis, 6.9% reported a history of incarceration, 11% had DR TB, 91% had 1 episode of TB treatment during the 2015 to 2019 period (i.e., diagnosed with TB only once), 7.9% had 2 episodes of TB treatment, and 1.6% had 3 to 8 episodes of TB treatment (Table 1).

The hepatitis C screening registry contained data on 1,849,820 adults tested for HCV antibodies between January 1, 2015 and September 30, 2020, who were not diagnosed with TB in that period. The median age in this population was 46 years (interquartile range [IQR]: 31 to 62), and 54.8% were male (Tables 2 and 3).

### Hepatitis C care cascade among persons with and without TB

After linking HCV testing information from the hepatitis C databases, 9,341 (78%) of 11,985 persons with TB were found to have been tested for HCV antibodies between January 1, 2015 and September 30, 2020 (Table 4). The proportion of patients with TB tested for HCV

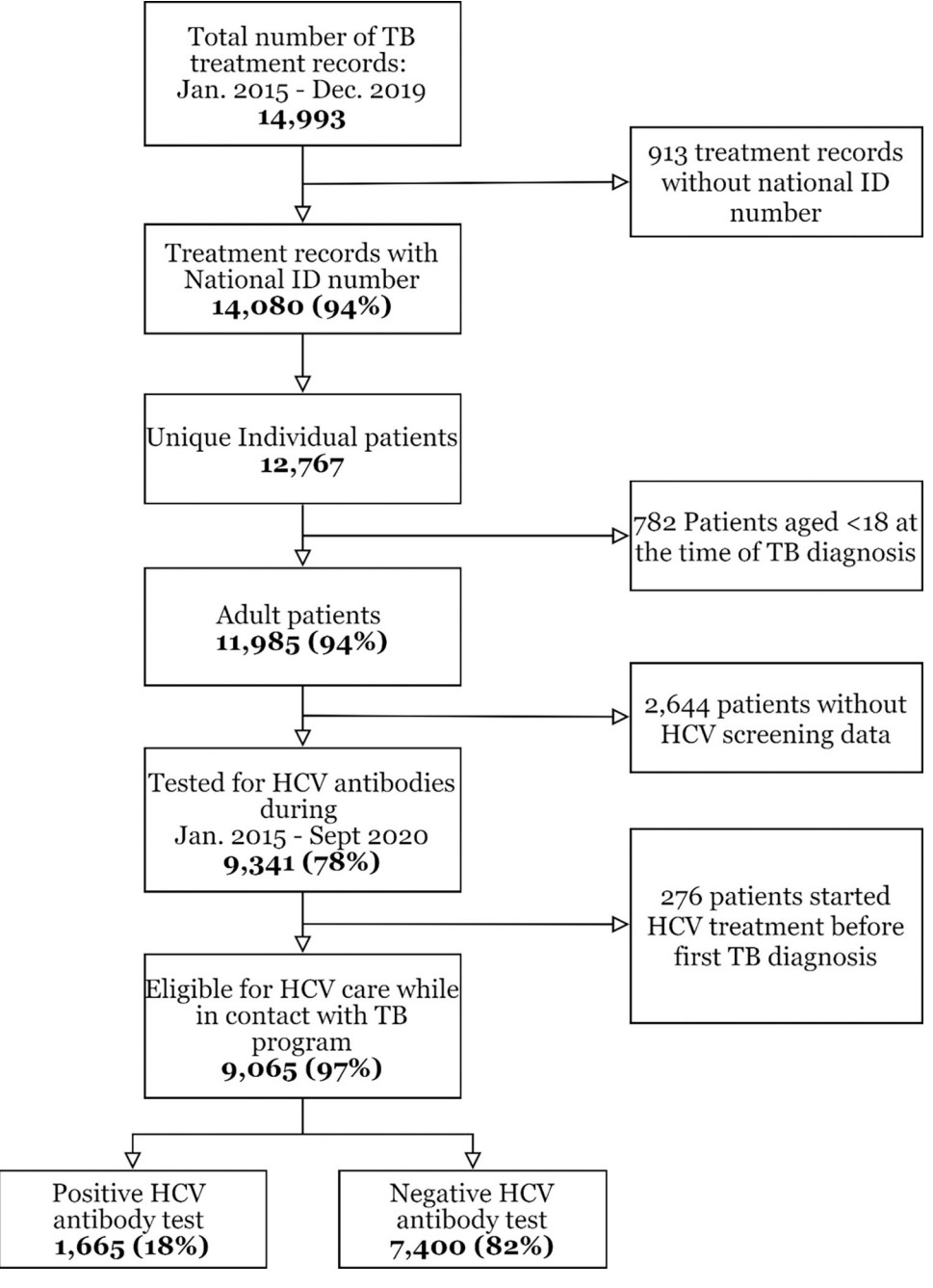

**Fig 2. Flow chart of patients with TB and their HCV screening status—2015–2019.** HCV, hepatitis C virus; TB, tuberculosis.

antibodies increased per year. Among patients diagnosed with TB in 2015, 60% were tested for HCV antibodies sometime during the study period. This proportion increased each subsequent year and reached 90% among patients diagnosed with TB in 2019. In the same period, HCV antibody-positivity in this population decreased from 25% among people diagnosed with TB in 2015 to 14% among those diagnosed in 2019. Among persons with TB included in our study, 276 (3.0%) had already started treatment for HCV infection before their first TB diagnosis and were included in the care cascade analysis of patients without TB.

**Table 4. HCV testing by year of first TB diagnosis among adult patients diagnosed with TB in 2015–2019.**

| Year | Patients diagnosed with TB | Ever tested for HCV antibodies[a] | | First HCV antibody testing before or during the last TB episode | | First HCV antibody testing after last TB episode outcome | | Ever tested positive for HCV antibodies | |
|---|---|---|---|---|---|---|---|---|---|
| | *N* | *N* | % | *N* | % (of total ever tested) | N | % (of total ever tested) | *N* | % |
| 2015 | 2,766 | 1,671 | 60% | 485 | 29% | 1,186 | 71% | 414 | 25% |
| 2016 | 2,740 | 1,860 | 68% | 855 | 46% | 1,005 | 54% | 410 | 22% |
| 2017 | 2,405 | 1,948 | 81% | 1,610 | 83% | 338 | 17% | 333 | 17% |
| 2018 | 2,086 | 1,791 | 86% | 1,635 | 91% | 156 | 9% | 255 | 14% |
| 2019 | 1,988 | 1,795 | 90% | 1,773 | 99% | 22 | 1% | 253 | 14% |

[a] Excluding patients who started treatment for HCV infection before first TB diagnosis.

HCV, hepatitis C virus; TB, tuberculosis.

Among the total of 9,065 patients with TB who were tested for hepatitis C, 1,665 (18%) patients (Fig 2) had a positive antibody test, of whom 108 (6%) died without undergoing viremia testing. Of the remaining 1,557 patients still alive, 316 (20%) were LTFU (i.e., never underwent viremia testing) (Figs 3A and S5); among 1,241 patients who underwent viremia results, viremic HCV infection was found in 1,025 (83%) patients, of whom 443 (43%) did not start hepatitis C treatment, including 80 persons that died (8% out of all viremic cases, 18% of those who did not start hepatitis C treatment). In a subset of patients with DR TB, the proportion of patients who were LTFU was even higher—27% and 52% after positive antibody and viremia tests, respectively (S6 Fig). For comparison, in the hepatitis C care cascade among patients without TB (Fig 3B), the proportion of patients who were LTFU from hepatitis C care after positive antibody test or after positive viremia test were 14% and 19%, respectively. Mortality was also lower among patients without TB: 3,842 (3%) of HCV antibody-positive persons died without viremia testing, and 3,127 (4%) viremic patients died without received hepatitis C treatment. Among patients initiating HCV treatment, those with TB were also less likely to complete hepatitis C treatment (89% versus 94%) and SVR assessment (59% versus 76%) than those without TB. However, among those who completed hepatitis C treatment and were tested for SVR, the cure rate was comparable to those without TB (98.3% versus 98.9%) (Figs 3A and 3B and S7 and S1 Table). Overall, among patients with confirmed viremic HCV infection, patients with TB were less likely to complete hepatitis C treatment (51%) and have a documented SVR achieved (28%), compared to patients without TB (76% and 55%, respectively; Fig 3C).

## Timeliness of HCV viremia testing and treatment initiation

Among those who had information on follow-up duration available, we used survival analysis methods to compare time from positive antibody test to viremia testing or censoring between patients with TB (*n* = 1,499) and patients without TB (*n* = 118,396). After positive HCV antibody test, 3,840 (3.2%) persons died, and patients without TB had viremia testing sooner than patients with TB (HR = 1.46, 95% CI [1.39, 1.54], *p* < 0.001) (Figs 4 and S8). Among those who underwent viremia testing, the median time from a positive antibody test result to viremia testing was 17 days (IQR: 3 to 248 days) among patients with TB (*n* = 1,116) and 6 days (IQR: 1 to 24) among patients without TB (*n* = 97,554) (*p* < 0.001).

We also compared time from positive viremia test to hepatitis C treatment initiation or censoring among 1,021 patients with TB and 87,964 patients without TB who had active viremia and information on follow-up duration available. After a positive viremia test, patients without

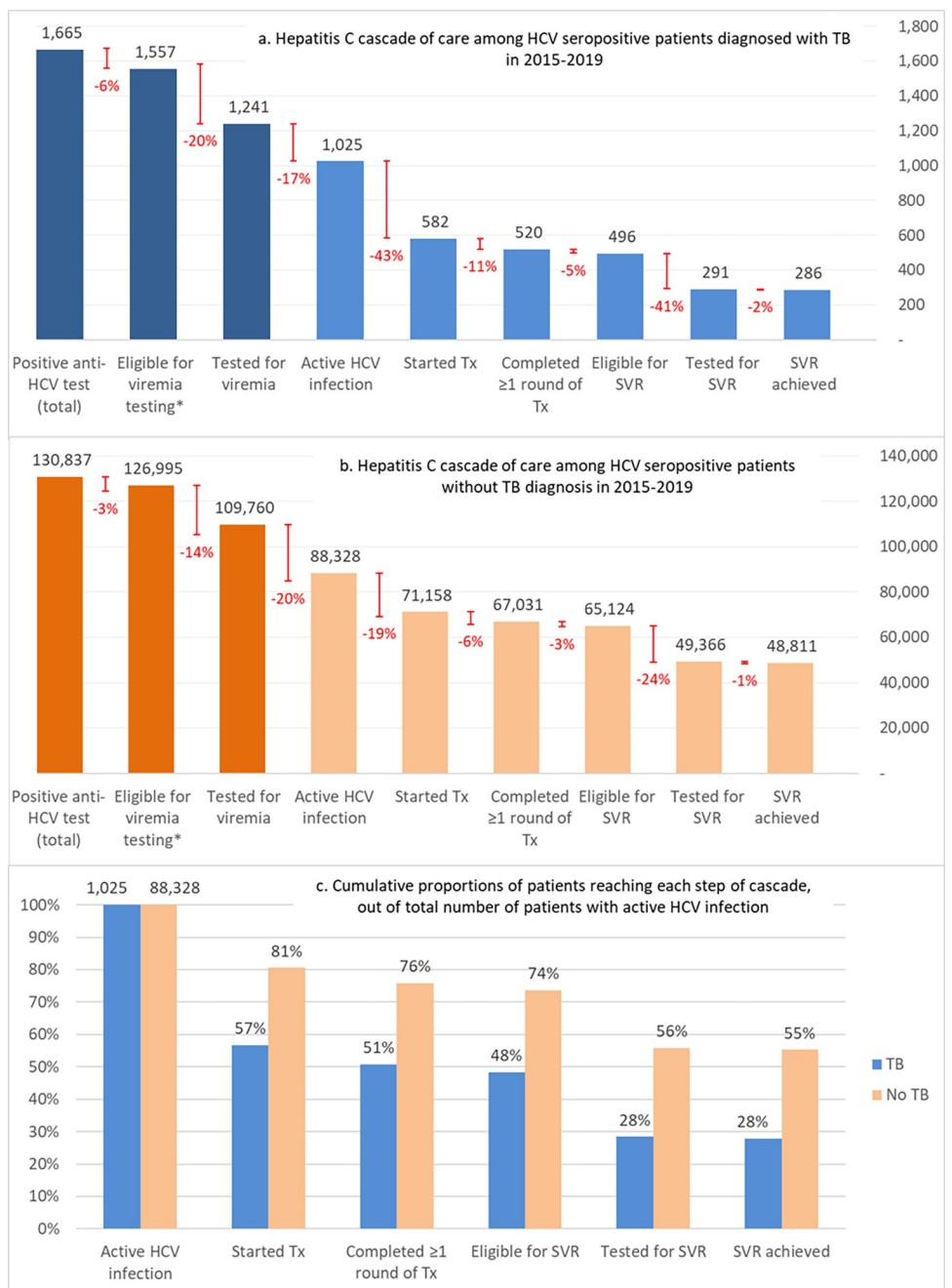

**Fig 3. Hepatitis C care cascade among patients with and without tuberculosis, Georgia, 2015–2020.** HCV, hepatitis C virus; SVR, sustained virologic response; TB, tuberculosis; Tx, treatment. Note: Red lines represent the percent change between 2 consecutive steps in the care cascade, i.e., adjacent bars of the charts.

TB started hepatitis C treatment sooner than patients with TB (HR = 2.05, 95% CI [1.87, 2.25], $p < 0.001$) (Figs 5 and S9). Among those who started hepatitis C treatment, the median time from confirmation to treatment initiation was 196 days (IQR: 76 to 523) among patients with MDR TB ($n = 78$), 99 days (IQR: 54 to 263) among patients with DS TB ($n = 460$), and 86 days (IQR: 56 to 203) among patients without TB ($n = 71,128$).

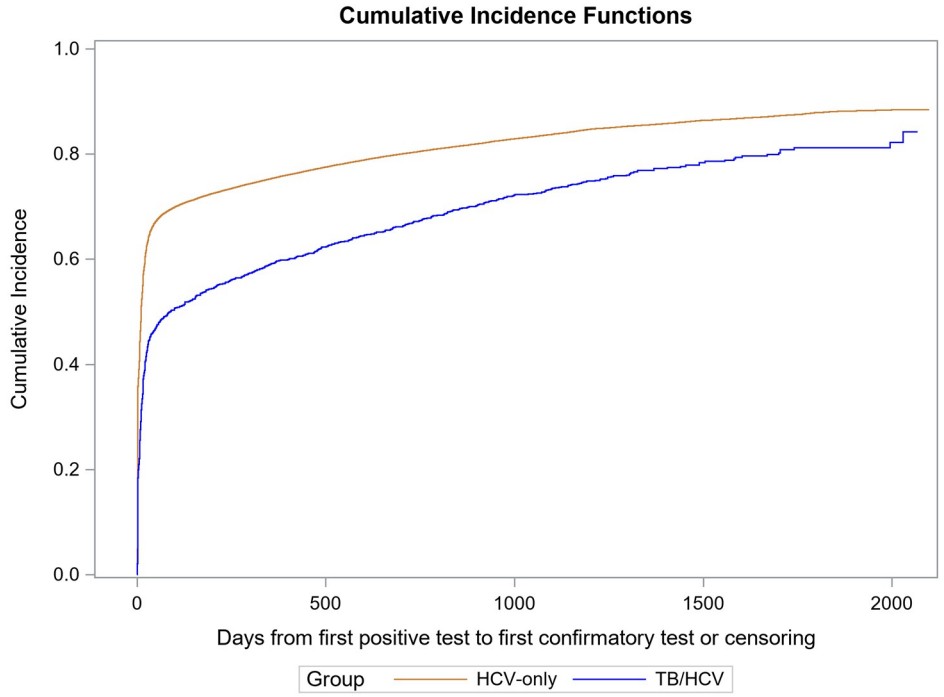

**Fig 4. Comparison of HCV viremia testing cumulative incidence among HCV seropositive patients with and without TB[a,b]: January 1, 2015–September 30, 2020.** [a] Blue line corresponds to the patients with confirmed HCV infection without evidence of TB diagnosis in 2015–2019. Red line corresponds to patients with confirmed HCV infection diagnosed with TB in 2015–2019. [b] Gray's test *p*-value <0.001. HCV, hepatitis C virus; TB, tuberculosis.

## Factors associated with LTFU from hepatitis C care among patients with TB

Among patients with TB who had a positive HCV antibody test result, LTFU before viremia testing was more common among females, patients enrolled in second-line TB treatment for drug-resistant TB, and patients with unsuccessful or unknown TB treatment outcomes (Table 5). LTFU at this step of the hepatitis C care cascade substantially decreased among patients diagnosed with TB in recent years, from 32% among patients diagnosed with TB in 2017 to 12% among patients diagnosed in 2019. In multivariable analysis, MDR TB was associated with increased risk of LTFU after positive HCV antibody test (aRR = 1.41, 95% CI [1.12, 1.76], *p* = 0.003).

Among patients with DS TB who had confirmed viremic HCV infection, LTFU before hepatitis C treatment initiation was more common among females, and patients with successful TB treatment outcomes (Table 6). The proportion of patients LTFU before hepatitis C treatment initiation increased by year, from 21% among patients diagnosed with TB in 2015 to 56% among patients diagnosed with TB in 2019. In multivariable analysis, we could not identify any factors meaningfully associated with LTFU before HCV treatment initiation.

## Discussion

In this study of 2 large-scale public health programs in the country of Georgia, we found that LTFU from the hepatitis C care cascade was high overall, but substantially more common among persons with TB than those without TB. Persons diagnosed with TB between 2015 and

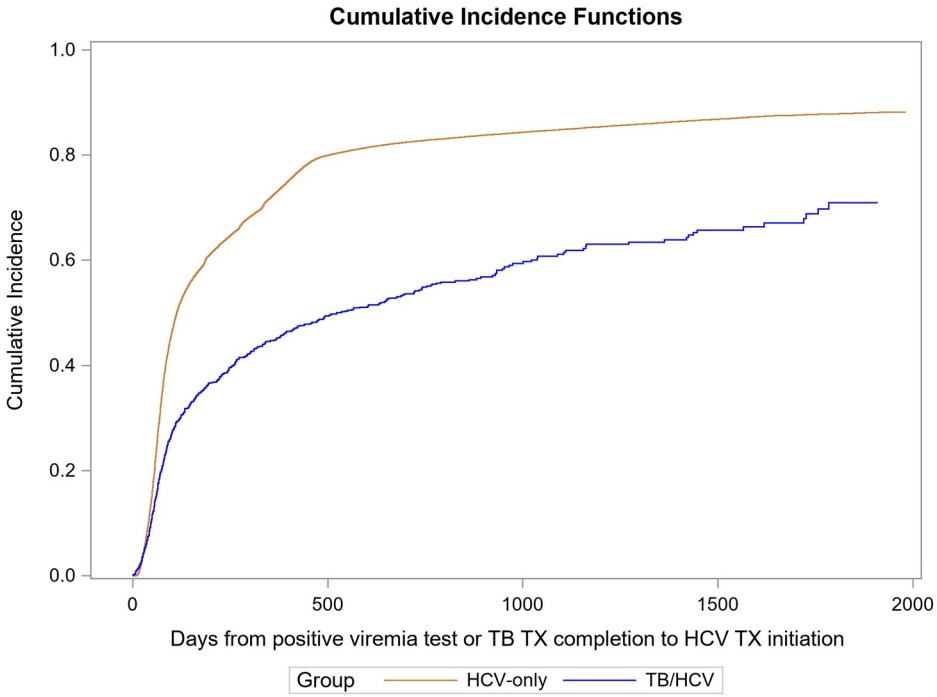

**Fig 5. Comparison of cumulative incidence of HCV treatment initiation among patients with confirmed HCV infection with and without TB[a,b]: January 1, 2015–September 30, 2020.** [a] Blue line corresponds to patients with confirmed HCV infection without evidence of TB diagnosis in 2015–2019. Red line corresponds to patients with confirmed HCV infection diagnosed with TB in 2015–2019. [b] Gray's test *p*-value <0.001. HCV, hepatitis C virus; TB, tuberculosis.

2019 were found to have a high prevalence (18%) of HCV antibodies, and approximately 20% of those with HCV antibodies did not receive viremia testing to evaluate for chronic HCV infection. Furthermore, almost 2 out of 5 patients with TB who had confirmed viremic HCV infection had not started hepatitis C treatment within the National Hepatitis C Elimination program in Georgia even after they completed TB treatment. We also found that proportion of patients with TB who get HCV antibody testing and subsequent viremia testing increased in recent years, while proportion of those with viremia who receive treatment decreased. Our findings highlight the importance of improving integration and linkage to hepatitis C diagnostic and treatment services among patients with TB.

LTFU at different stages of hepatitis C care is a major barrier in controlling HCV infection globally [17]. Timely treatment of hepatitis C reduces the risk of cirrhosis, hepatocellular carcinoma, and death [37,38]. Therefore, LTFU from hepatitis C care can increase the risk of adverse health outcomes. Additionally, untreated hepatitis C contributes to further transmission of the infection, hindering progress towards achieving elimination goals globally [17]. Georgia faces the same problem despite the nationwide Hepatitis C Elimination Program, which offers free diagnostic and treatment services. Previous studies from Georgia have found that a high proportion of patients are LTFU before viremia testing and before hepatitis C treatment initiation [26,28]. In the first few years of the program, some patients had to co-pay for the viremia testing and other pretreatment diagnostic procedures, which was one of the main barriers for enrollment in the treatment [39,40]. Since then, this barrier has been removed by eliminating any copayment obligation for patients [28], but the LTFU still remains a challenge. Recent efforts that aimed to re-engage the persons who are LTFU in the care by actively

**Table 5. LTFU from HCV care before HCV viremia testing among patients with TB.**

| Characteristics (based on first TB diagnosis) | Total (N = 1,557) | | LTFU (n = 318) | | Not LTFU (n = 1,239) | | cRR (95% CI) | P-value | aRR (95% CI) | P-value |
|---|---|---|---|---|---|---|---|---|---|---|
| | N | % (col) | N | % (row) | N | % (row) | | | | |
| **Sex** | | | | | | | | | | |
| Male | 1,411 | 91% | 273 | 19% | 1,138 | 81% | 1 | | | |
| Female | 146 | 9% | 45 | 31% | 101 | 69% | 1.59 (1.22, 2.08) | <0.001 | | |
| **Age** | | | | | | | | | | |
| Median (IQR) | 45 (16) | | 43 (16) | | 46 (16) | | | <0.001 | | |
| **Location** | | | | | | | | | | |
| Tbilisi | 620 | 40% | 138 | 22% | 482 | 78% | 1 | | | |
| Rest of Georgia | 885 | 57% | 173 | 20% | 712 | 80% | 0.88 (0.72, 1.07) | 0.200 | | |
| Penitentiary system | 52 | 3% | 7 | 13% | 45 | 87% | 0.60 (0.30, 1.22) | 0.162 | | |
| **Employment status** | | | | | | | | | | |
| Employed | 155 | 10% | 34 | 22% | 121 | 78% | 1 | | 1 | |
| Unemployed | 1,344 | 86% | 276 | 21% | 1,068 | 79% | 0.94 (0.68, 1.28) | 0.682 | 0.96 (0.71, 1.31)[a] | 0.815 |
| Military | 2 | 0% | 0 | 0% | 2 | 100% | - | | | |
| Missing | 56 | 4% | 8 | 14% | 48 | 86% | - | | | |
| **IDP** | | | | | | | | | | |
| Yes | 131 | 8% | 37 | 28% | 94 | 72% | 1.44 (1.07, 1.93) | 0.015 | | |
| No | 1,342 | 86% | 263 | 20% | 1,079 | 80% | 1 | | | |
| Missing | 84 | 5% | 18 | 21% | 66 | 79% | - | | | |
| **History of imprisonment** | | | | | | | | | | |
| Yes | 295 | 19% | 55 | 19% | 240 | 81% | 0.91 (0.70, 1.18) | 0.468 | | |
| No | 1,192 | 77% | 245 | 21% | 947 | 79% | 1 | | | |
| Missing | 70 | 4% | 18 | 26% | 52 | 74% | - | | | |
| **TB diagnosis and treatment** | | | | | | | | | | |
| **Year of TB diagnosis** | | | | | | | | | | |
| 2015 | 386 | 25% | 64 | 17% | 322 | 83% | 1.35 (0.90, 2.02) | 0.146 | | |
| 2016 | 374 | 24% | 76 | 20% | 298 | 80% | 1.65 (1.12, 2.44) | 0.012 | | |
| 2017 | 304 | 20% | 97 | 32% | 207 | 68% | 2.60 (1.79, 3.77) | <0.001 | | |
| 2018 | 249 | 16% | 51 | 20% | 198 | 80% | 1.67 (1.10, 2.52) | 0.016 | | |
| 2019 | 244 | 16% | 30 | 12% | 214 | 88% | 1 | | | |
| **Newly diagnosed TB** | | | | | | | | | | |
| Yes | 1,066 | 68% | 217 | 20% | 849 | 80% | 1 | | 1 | |
| No/Unknown | 491 | 32% | 101 | 21% | 390 | 79% | 1.01 (0.82, 1.25) | 0.923 | 1.06 (0.85, 1.33)[b] | 0.605 |
| **MDR TB** | | | | | | | | | | |
| Yes | 277 | 18% | 75 | 27% | 202 | 73% | 1.43 (1.14, 1.78) | 0.002 | 1.41 (1.12, 1.76)[c] | 0.003 |
| No | 1,280 | 82% | 243 | 19% | 1037 | 81% | 1 | | 1 | |
| **Treatment outcome** | | | | | | | | | | |
| Successful | 985 | 63% | 184 | 19% | 801 | 81% | 1 | | 1 | |
| Unsuccessful/Unknown | 509 | 33% | 125 | 25% | 384 | 75% | 1.31 (1.08, 1.61) | 0.008 | 1.13 (0.87, 1.49)[d] | 0.361 |
| Missing | 63 | 4% | 9 | 14% | 54 | 86% | - | | | |

Study population in this analysis included patients diagnosed with TB in 2015–2019 who had a positive HCV antibody test result. Comparison groups: (1) Patients with TB who did not undergo viremia testing and are still alive (LTFU); and (2) patients with TB who underwent HCV viremia testing (not LTFU). HCV viremia testing status ascertained as of September 30, 2020. Patients with positive HCV antibody test who died are excluded from this analysis. *P*-value was calculated using chi-square test for all variables, except the age comparison where Wilcoxon rank-sum test was used.

[a] Adjusted for age, sex, IDP, case definition (new vs. previously treated).

[b] Adjusted for age, sex, MDR TB, employment status.

[c] Adjusted for age, sex, case definition (new vs. previously treated).

[d] Adjusted for age, sex, MDR TB, employment status.

aRR, adjusted risk ratio; cRR, crude risk ratio; HCV, hepatitis C virus; IDP, internally displaced person; IQR, interquartile range; LTFU, loss to follow-up; MDR, multidrug-resistant TB; TB, tuberculosis; Tx, treatment.

**Table 6. LTFU from hepatitis C care before treatment initiation among patients with drug-susceptible TB.**

| Characteristics (based on first TB diagnosis) | Total (N = 758) | | LTFU (n = 259) | | Not LTFU (n = 499) | | cRR (95% CI) | P-value | aRR (95% CI) | P-value |
|---|---|---|---|---|---|---|---|---|---|---|
| | N | % | N | % | N | % | | | | |
| **Demographic characteristics** | | | | | | | | | | |
| **Sex** | | | | | | | | | | |
| Male | 694 | 92% | 230 | 33% | 464 | 67% | 1 | | | |
| Female | 64 | 8% | 29 | 45% | 35 | 55% | 1.37 (1.02, 1.83) | 0.034 | | |
| **Age** | | | | | | | | | | |
| Median (IQR) | 45 (15) | | 46 (16) | | 44 (14) | | | 0.015 | | |
| **Location** | | | | | | | | | | |
| Tbilisi | 290 | 38% | 95 | 33% | 195 | 67% | 1 | | | |
| Rest of Georgia | 441 | 58% | 157 | 36% | 284 | 64% | 1.09 (0.88, 1.34) | 0.431 | | |
| Penitentiary system | 27 | 4% | 7 | 26% | 20 | 74% | 0.79 (0.41, 1.53) | 0.486 | | |
| **Employment status** | | | | | | | | | | |
| Employed | 75 | 10% | 25 | 33% | 50 | 67% | 1 | | | |
| Unemployed | 647 | 85% | 218 | 34% | 429 | 66% | 1.01 (0.72, 1.42) | 0.950 | 1.01 (0.71, 1.41)[a] | 0.969 |
| Military | 2 | 0% | 1 | 50% | 1 | 50% | - | | | |
| Missing | 34 | 4% | 15 | 44% | 19 | 56% | - | | | |
| **IDP** | | | | | | | | | | |
| Yes | 67 | 9% | 22 | 33% | 45 | 67% | 0.97 (0.68, 1.39) | 0.882 | | |
| No | 655 | 86% | 221 | 34% | 434 | 66% | 1 | | | |
| Missing | 36 | 5% | 16 | 44% | 20 | 56% | | | | |
| **History of imprisonment** | | | | | | | | | | |
| Yes | 597 | 79% | 47 | 36% | 84 | 64% | 1.05 (0.81, 1.35) | 0.707 | | |
| No | 131 | 17% | 204 | 34% | 393 | 66% | 1 | | | |
| Missing | 30 | 4% | 8 | 27% | 22 | 73% | | | | |
| **TB diagnosis and treatment** | | | | | | | | | | |
| **Year of TB diagnosis** | | | | | | | | | | |
| 2015 | 193 | 25% | 41 | 21% | 152 | 79% | 1 | | | |
| 2016 | 199 | 26% | 46 | 23% | 153 | 77% | 1.09 (0.75, 1.58) | 0.656 | | |
| 2017 | 135 | 18% | 47 | 35% | 88 | 65% | 1.64 (1.15, 2.34) | 0.007 | | |
| 2018 | 117 | 15% | 61 | 52% | 56 | 48% | 2.45 (1.78, 3.39) | <0.001 | | |
| 2019 | 114 | 15% | 64 | 56% | 50 | 44% | 2.64 (1.93, 3.63) | <0.001 | | |
| **# of TB Tx episodes (binary)** | | | | | | | | | | |
| 1 | 647 | 85% | 231 | 36% | 416 | 64% | 1 | | | |
| 2+ | 111 | 15% | 28 | 25% | 83 | 75% | 0.71 (0.50, 0.99) | 0.043 | | |
| **Newly diagnosed TB** | | | | | | | | | | |
| Yes | 555 | 73% | 193 | 35% | 362 | 65% | 1 | | 1 | |
| No/Unknown | 203 | 27% | 66 | 33% | 137 | 67% | 0.93 (0.74, 1.18) | 0.564 | 0.91 (0.72, 1.15)[b] | 0.441 |
| **Treatment outcome (binary)** | | | | | | | | | | |
| Successful | 611 | 81% | 214 | 35% | 397 | 65% | 1.09 (0.84, 1.42) | 0.523 | 1.04 (0.79, 1.36)[c] | 0.792 |
| Unsuccessful/Unknown | 137 | 18% | 44 | 32% | 93 | 68% | 1 | | 1 | |
| Missing | 10 | 1% | 1 | 10% | 9 | 90% | - | | | |

Study population included in this table include patients diagnosed with drug-susceptible TB in 2015–2019 who have confirmed HCV infection (positive viremia test). Comparison groups: (1) Patients with TB who are eligible but did not start HCV treatment as of September 2020 (LTFU); and (2) patients with TB who started HCV treatment (not LTFU). Patients with positive HCV viremia test who died or are still on TB treatment are excluded from this analysis. P-value was calculated using chi-square test for all variables, except the age comparison where Wilcoxon rank-sum test was used.

[a] Adjusted for age, sex, IDP, case definition (new vs. previously treated).

[b] Adjusted for age, sex, employment status.

[c] Adjusted for age, sex, employment status.

aRR, adjusted risk ratio; cRR, crude risk ratio; HCV, hepatitis C virus; IDP, internally displaced person; IQR, interquartile range; LTFU, loss to follow-up; TB, tuberculosis; Tx, treatment.

contacting them achieved some success [41], but the gap is still wide. Similar interventions to promote linkage to care among specific target groups, such as patients with TB, have not been put in place. Our findings demonstrate LTFU from hepatitis C care is even more pronounced among patients with active TB disease coinfected with HCV. Notably, among patients with TB, LTFU before HCV viremia testing has decreased in recent years, which could be explained by a change in policy. According to Georgian 2018 TB management guideline, if an HCV antibody test is positive, blood samples are taken from the patient and sent for HCV viremia testing. Therefore, this policy change, coupled with complete financial coverage of viremia testing by the government [28], might have removed geographic, logistical, and financial barriers to viremia testing for patients with TB. This improvement highlights that systemic changes such as financial and logistic support could be key to improving retention of patients in the hepatitis C care.

Furthermore, LTFU before hepatitis C treatment initiation is also very common among patients with TB. Higher LTFU before hepatitis C treatment initiation among patients with TB compared to those without TB could be explained by the fact that TB treatment is long (from 6 to 24 months) and can be associated with severe adverse reactions [42]. Additionally, HCV treatment is usually not initiated until TB treatment completion due to drug–drug interaction concerns [20]. Therefore, in the absence of a formal linkage program, patients with TB might be more reluctant to start another long treatment course for hepatitis C due to treatment fatigue and previous negative experiences [43]. This issue has not been explicitly studied among patients with both TB and HCV. Still, fatigue from diagnostic and treatment procedures as a risk factor for treatment discontinuation has been described among patients with TB and those coinfected with human immunodeficiency virus (HIV) [44–47]. This is even more likely in the case of HCV infection without advanced liver damage because patients might not experience any symptoms associated with HCV infection and might not feel the urgency to seek hepatitis C care [48].

We also found that the proportion of patients LTFU before hepatitis C treatment initiation is increasing by year among patients who completed treatment for DS TB. Several reasons could explain this finding: (1) Patients with TB diagnosed in recent years had less time to get enrolled in HCV elimination program after completing their TB treatment; (2) the COVID-19 pandemic and related restrictions in 2020 could have affected patients' ability and willingness to start treatment for hepatitis C; and (3) the change in policy in 2018, as described above, removed a barrier to viremia testing for many patients, increasing the denominator of patients with TB diagnosed with HCV infection and eligible for treatment. This policy change that increased the viremia testing rates was not accompanied with a similar intervention to increase treatment initiation rates. Therefore, we can speculate that the number of patients who needed treatment (i.e., those with positive viremia result) increased at a higher rate than the number of patients who received treatment, reflected in increasing proportion of LTFU before treatment initiation in recent years. Additional surveys should be conducted among patients with TB to identify more specific reasons and challenges for not initiating hepatitis C treatment after a positive viremia test.

Overall, hepatitis C care cascade analyses in other countries show a substantial heterogeneity in terms of viremia testing and treatment uptake. However, LTFU after positive antibody or viremia test is a universal challenge that many countries face, with Georgia usually having higher than average rates. For example, 1 review found that among countries reporting the nationwide data, Georgia had the highest rate of viremia testing among HCV antibody positive individuals [49]. Treatment uptake among HCV viremia–positive individuals ranged even more widely, from 5% to 95%. Georgia with 80% overall treatment uptake was ranked higher on this spectrum.

Due to these similarities of challenges in hepatitis C care cascade across countries, results from this study have potential implications for other countries with national TB programs. In 2015, Georgia became the first country to formally initiate a nationwide hepatitis C elimination program, with significant progress achieved in scaling up HCV screening, diagnostic and treatment services throughout the country [26]. However, previously published literature and our analysis highlight that LTFU from different stages of hepatitis C care poses a major challenge to HCV elimination goals, and patients with TB are even more likely to be LTFU from hepatitis C care. To address this issue, program partners should consider introducing additional activities directed at more integrated care of hepatitis C and TB and interventions targeted at retaining patients in hepatitis C care. These interventions could include patient navigators that have been found effective in hepatitis C linkage to care [50] or providing hepatitis C treatment in the same facilities where they receive TB treatment. Additionally, patients with MDR TB could be treated for HCV infection and TB concomitantly. There is no documented drug–drug interaction between DAAs used in HCV treatment and second-line TB drugs [20], and concomitant treatment of TB and HCV infection have been successfully implemented in small samples of patients [51,52]. For patients with DS TB, replacing rifampin with rifabutin may allow for simultaneous TB and HCV treatment [20]. Therefore, the existing infrastructure of NTP can help the HCV elimination efforts in Georgia by achieving microelimination of HCV infection among patients with TB—a strategy that is proposed as a useful approach for achieving the overall elimination goals [53].

In addition to tuberculosis, there are several other conditions that have substantial overlap with hepatitis C, such as HIV infection, injection drug use, and incarceration. Georgia already has a good experience of integrating services for some of these different conditions. For example, Georgia is considered as an example of good practice in terms of integrating the diagnostic services for HIV, tuberculosis, and hepatitis C in primary healthcare [54]. Hepatitis C testing and treatment in corrections system started even before the launch of the nationwide hepatitis C elimination program [55]. Furthermore, hepatitis C treatment for people living with HIV is usually offered in the same facilities where they receive antiretroviral treatment. However, hepatitis C viremia testing and treatment is not systematically integrated in the harm reduction sites, even though it has been found feasible and preferable for people who inject drugs to receive these services in the same sites [56,57].

Our study has several limitations. First, due to missing national ID numbers, we had to exclude 6% of observations from the NTP database. Second, the limited number of variables in the hepatitis C screening registry and differences in variables available in hepatitis C and TB databases did not allow us to conduct a more in-depth analysis to adjust for potential confounders. For that reason, our time-to-event analyses are limited to the crude comparison of cumulative incidence curves and crude hazards ratios between patients with and without TB, rather than trying to explore any causal associations that would require confounding adjustment. However, even without exploring the causal association, the pronounced difference between patients with both TB and hepatitis C and those with only hepatitis C warrants the attention to reduce the gap in linkage to hepatitis C care and treatment among patients with TB. Third, we excluded patients with DR TB from the risk factor analysis of LTFU before HCV treatment initiation due to high heterogeneity in terms of when individual patients with DR TB become eligible for hepatitis C treatment.

In conclusion, we found that LTFU from hepatitis C care after positive HCV antibody and viremia testing is more common among patients with TB than those without TB. Existing large-scale public health programs for both TB and hepatitis C in Georgia create a unique opportunity for integrated care of these 2 infectious diseases, which could potentially reduce LTFU. Though our study was not designed to identify effective interventions for reducing

LTFU, integrated care should utilize a patient-centered approach and include the following suggested interventions: (1) merging care for TB and HCV infection for coinfected patients; (2) parallel treatment of TB and hepatitis C using drug regimens without substantial drug–drug interactions; (3) active support for patients with TB to navigate the referral process and enroll in the hepatitis C treatment program, whenever it is not feasible to receive HCV care at the same facility where they receive TB treatment. These interventions can positively impact both TB and hepatitis C programs by providing timely hepatitis C care to patients with TB, thereby reducing LTFU and improving overall patient health outcomes.

## Supporting information

**S1 Checklist. STROBE Statement—Checklist of items that should be included in reports of cohort studies.**
(DOCX)

**S2 Checklist. Inclusivity in global research.**
(DOCX)

**S1 Analysis Plan. Analysis plan.**
(DOCX)

**S1 Fig. Directed acyclic graph depicting the causal relations between exposure of interest (Employment), outcome of interest (loss to follow-up), and other covariates. IDP, internally displaced person; TB, tuberculosis.**
(DOCX)

**S2 Fig. Directed acyclic graph depicting the causal relations between exposure of interest (MDR TB), outcome of interest (loss to follow-up), and other covariates. MDR, multidrug-resistant; TB, tuberculosis.**
(DOCX)

**S3 Fig. Directed acyclic graph depicting the causal relations between exposure of interest (previously treated TB), outcome of interest (loss to follow-up), and other covariates. MDR, multidrug-resistant; TB, tuberculosis.**
(DOCX)

**S4 Fig. Directed acyclic graph depicting the causal relations between exposure of interest (TB treatment outcome), outcome of interest (loss to follow-up), and other covariates. MDR, multidrug-resistant; TB, tuberculosis.**
(DOCX)

**S5 Fig. Care cascade of hepatitis C among patients with tuberculosis, starting from the total number of patients with TB.** HCV, hepatitis C virus; SVR, sustained virologic response; TB, tuberculosis; Tx, treatment.
(DOCX)

**S6 Fig. Hepatitis C care cascade among HCV seropositive patients with TB, stratified by their drug-resistance status.** Note: Red lines represent the percent change between 2 consecutive steps in the care cascade, i.e., adjacent bars of the charts. HCV, hepatitis C virus; SVR, sustained virologic response; TB, tuberculosis; Tx, treatment.
(DOCX)

**S7 Fig. Comparison of hepatitis C care cascade among patients with and without TB in Georgia, 2015–2020.** HCV, hepatitis C virus; SVR, sustained virologic response; TB,

tuberculosis; Tx, treatment.
(DOCX)

**S8 Fig. Kaplan–Meier curves of time from positive antibody test to viremia testing, with 95% confidence bands.** HCV, hepatitis C virus; TB, tuberculosis.
(DOCX)

**S9 Fig. Kaplan–Meier curves of time from positive viremia test or TB treatment completion (whichever occurred last) to hepatitis C treatment initiation, with 95% confidence bands.** HCV, hepatitis C virus; TB, tuberculosis.
(DOCX)

**S1 Table. Comparison of proportions at each step of care cascade between patients with and without TB, Georgia, 2015–2020.** HCV, hepatitis C virus; SVR, sustained virologic response; TB, tuberculosis; Tx, treatment.
(DOCX)

## Acknowledgments

The authors would like to thank Amiran Gamkrelidze and Zaza Avaliani for their support in obtaining the data used in this study.

## Disclaimer

The findings and conclusions in this report are those of the authors and do not necessarily represent the official position of the US Centers for Disease Control and Prevention.

## Author Contributions

**Conceptualization:** Davit Baliashvili, Henry M. Blumberg, Neel R. Gandhi, Shaun Shadaker, Lia Gvinjilia, Nestani Tukvadze, Maia Butsashvili, Russell R. Kempker.

**Data curation:** Davit Baliashvili, Shaun Shadaker, Lia Gvinjilia, Aleksandre Turdziladze, Nestani Tukvadze, Mamuka Chincharauli, Maia Butsashvili, Lali Sharvadze, Tengiz Tsertsvadze, Jaba Zarkua.

**Formal analysis:** Davit Baliashvili, David Benkeser.

**Funding acquisition:** Henry M. Blumberg, Neel R. Gandhi, Russell R. Kempker.

**Investigation:** Russell R. Kempker.

**Methodology:** Davit Baliashvili, Henry M. Blumberg, Neel R. Gandhi, Francisco Averhoff, David Benkeser, Shaun Shadaker, Lia Gvinjilia, Nestani Tukvadze, Maia Butsashvili, Lali Sharvadze, Tengiz Tsertsvadze, Jaba Zarkua, Russell R. Kempker.

**Project administration:** Davit Baliashvili, Aleksandre Turdziladze, Russell R. Kempker.

**Resources:** Aleksandre Turdziladze, Nestani Tukvadze, Mamuka Chincharauli, Lali Sharvadze, Tengiz Tsertsvadze, Jaba Zarkua.

**Software:** Davit Baliashvili, Shaun Shadaker, Aleksandre Turdziladze, Mamuka Chincharauli.

**Supervision:** Henry M. Blumberg, Neel R. Gandhi, Francisco Averhoff, David Benkeser, Shaun Shadaker, Lia Gvinjilia, Russell R. Kempker.

**Validation:** Davit Baliashvili, Russell R. Kempker.

**Visualization:** Davit Baliashvili.

**Writing – original draft:** Davit Baliashvili.

**Writing – review & editing:** Davit Baliashvili, Henry M. Blumberg, Neel R. Gandhi, Francisco Averhoff, David Benkeser, Shaun Shadaker, Lia Gvinjilia, Aleksandre Turdziladze, Nestani Tukvadze, Mamuka Chincharauli, Maia Butsashvili, Lali Sharvadze, Tengiz Tsertsvadze, Jaba Zarkua, Russell R. Kempker.

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
