## [Editor Report · Decision Letter 0]

7 Oct 2022

Dear Dr Baliashvili, 

Thank you for submitting your manuscript entitled "Hepatitis C care cascade among patients with and without tuberculosis: findings from nationwide programs in the country of Georgia, 2015-2020" for consideration by PLOS Medicine.

Your manuscript has now been evaluated by the PLOS Medicine editorial staff as well as by an academic editor with relevant expertise and I am writing to let you know that we would like to send your submission out for external peer review.

Please re-submit your manuscript within two working days, i.e. by Oct 11 2022 11:59PM.

Kind regards,

Philippa Dodd, MBBS MRCP PhD

Senior Editor

PLOS Medicine

---

## [Decision Letter · Decision Letter 1]

12 Dec 2022

Dear Dr. Baliashvili,

Thank you very much for submitting your manuscript "Hepatitis C care cascade among patients with and without tuberculosis: findings from nationwide programs in the country of Georgia, 2015-2020" (PMEDICINE-D-22-03278R1) for consideration at PLOS Medicine. 

[LINK]

In light of these reviews, I am afraid that we will not be able to accept the manuscript for publication in the journal in its current form, but we would like to consider a revised version that addresses the reviewers' and editors' comments. Obviously we cannot make any decision about publication until we have seen the revised manuscript and your response, and we plan to seek re-review by one or more of the reviewers. 

We expect to receive your revised manuscript by Jan 02 2023 11:59PM. Please email us (plosmedicine@plos.org) if you have any questions or concerns.

We look forward to receiving your revised manuscript. 

Sincerely,

Philippa Dodd, MBBS MRCP PhD

PLOS Medicine

plosmedicine.org

GENERAL

Please respond to all editor and reviewer comments detailed below in full

Please ensure that the study is reported according to the STROBE guideline, and include the completed STROBE checklist as Supporting Information. Please add the following statement, or similar, to the Methods: "This study is reported as per the Strengthening the Reporting of Observational Studies in Epidemiology (STROBE) guideline (S1 Checklist)."

COMMENTS FROM THE ACADEMIC EDITOR

There have been several papers looking at HCV treatment cascades, although most come from Europe/US among selected populations or among other target populations (IVDU and/or incarcerated persons). Although TB-positive patients are perhaps less relevant to US and parts of Europe, this is important from international perspective and several countries with remaining HCV pockets. With that in mind, I agree reasonable to proceed with major revision. In addition to comments from reviewers, I had some thoughts I listed below:

1. Greater details regarding any interventions that were put in place to promote linkage to care (even as simple as local or national education campaigns).

2. There was improvement in HCV Ab testing over time; was there changes in follow-up testing over time?

3. Patients with active HCV who failed to get treated could use more detail. Were these patients referred for care and/or seen in clinic?

4. Do authors have any data regarding comorbidity or life expectancy of patients? Some of the cascade failures may have been appropriate lack of referrals/treatment given comorbidities, including decompensated cirrhosis (Child C cirrhosis) where HCV treatment is expected to have less impact.

5. The TB and non-TB populations are inherently different but I did not see characteristics of the patients, stratified by TB status to detail those differences. To that end, I question if a propensity-score matched analysis would be possible/informative.

TITLE

Please revise your title according to PLOS Medicine's style. Your title must be nondeclarative and not a question. It should begin with main concept if possible. "Effect of" should be used only if causality can be inferred, i.e., for an RCT. Please place the study design ("A randomized controlled trial," "A retrospective study," "A modelling study," etc.) in the subtitle (ie, after a colon).

ABSTRACT

Please structure your abstract using the PLOS Medicine headings (Background, Methods and Findings, Conclusions).

Please combine the Methods and Findings sections into one section, “Methods and findings”.

Abstract Background:

Line 122 of your main introduction details the objectives of your study slightly different to those presented in the abstract. Please amend the abstract accordingly for consistency, clarity and to better inform the reader of the abstract. 

Abstract Methods and Findings:

Please ensure that all numbers presented in the abstract are present and identical to numbers presented in the main manuscript text.

Please elaborate on the details of the population studied and the setting, as you have done in the main methods section. 

Please quantify the main results for the outcomes measured with 95% CIs and p values as well as hazard ratios and the risk estimates that you describe in the main methods section. 

Please ensure that you include the actual amounts and/or absolute risk(s) of relevant outcomes (including NNT or NNH where appropriate), not just relative risks or correlation coefficients. (example for absolute risks: PMID: 28399126). 

Please include the important dependent variables that are adjusted for in the analyses.

In the last sentence of the Abstract Methods and Findings section, please describe the main limitation(s) of the study's methodology.

Abstract Conclusions:

Does the study have impact outside of Georgia? If so, it would be helpful to include this

AUTHOR SUMMARY

At this stage, we ask that you include a short, non-technical Author Summary of your research to make findings accessible to a wide audience that includes both scientists and non-scientists. The Author Summary should immediately follow the Abstract in your revised manuscript. This text is subject to editorial change and should be distinct from the scientific abstract. Please see our author guidelines for more information: 

https://journals.plos.org/plosmedicine/s/revising-your-manuscript#loc-author-summary

METHODS and RESULTS

Did your study have a prospective protocol or analysis plan? Please state this (either way) early in the Methods section.

For all observational studies, we ask that in the manuscript text, you please indicate: 

(1) the specific hypotheses you intended to test, 

(2) the analytical methods by which you planned to test them, 

(3) the analyses you actually performed, and 

(4) when reported analyses differ from those that were planned, transparent explanations for differences that affect the reliability of the study's results. If a reported analysis was performed based on an interesting but unanticipated pattern in the data, please be clear that the analysis was data-driven.

Where you report 95% CIs, please also report p-values and the statistical test used to determine them. Suggest reporting as follows, for example, line 266: “…with TB (HR = 1.46, 95% CI: 1.39, 1.54)…” suggest reporting as “(HR: 1.46, 95% CI [1.39, 1.54], p< or =) please specify the significance level i.e. 0.05, 0.01. Please amend throughout the main manuscript text and the abstract and the author summary.

TABLES and FIGURES

We agree with the reviewer than inclusion of the DAG would be helpful. Please see below and include as a supplementary file

We ask all authors to provide a separate table showing the complete baseline characteristics of the study population, please include

Please ensure all figures/tables have appropriate captions describing their contents and defining any and all abbreviations including those used for statistical reporting including the supplementary files. 

Figure 1: please ensure all numerical values are identical to those presented in the manuscript text

Figure 2: please see statistical reviewer comments below, which we agree with. It is unclear what all the numerical values on this figure refer to. Please revise accordingly. Please clearly define the meaning of the bars and whiskers in an appropriate caption. Results should be quantified using p-values and 95% CIs. Please apply the same revisions to the figures in the supplementary files which are presented in the same manner (figure S2)

Table 2: we agree with the reviewer that a label other than “other” would be preferable, if possible, when describing regions. Where you report 95% CIs please also report p-values in a separate column and in the table caption please define the statistical test used to determine them. 

DISCUSSION

Line 308: “globally.[17]” please ensure in-text reference callouts are placed before punctuation in square parentheses: as follows “globally [17].” Please check and amend throughout.

Line 315: “[26, 28]” please remove spaced between citations as follows “[26,28]”

REFERENCES

Please see our guidelines for referencing here: https://journals.plos.org/plosmedicine/s/submission-guidelines#loc-references

Journal name abbreviations should be those found in the National Center for Biotechnology Information (NCBI) databases. 

In the bibliography please ensure that up to but no more than 6 authors are listed followed by et al, where more than 6 authors contribute to the study

PARACHUTE RESEARCH POLICY

We note that you conducted research or obtained samples in a foreign country. Did you consider including a local author as first or last author? If not, we recommend that you consider doing so in line with ICMJE's authorship requirements (https://www.icmje.org/recommendations/browse/roles-and-responsibilities/defining-the-role-of-authors-and-contributors.html). PLOS has a parachute research policy which aims to promote collaboration and inclusivity in global health research. You are required to complete PLOS’ questionnaire on inclusivity in global research and submit it with your revised paper. The policy and questionnaire can be found at https://journals.plos.org/plosone/s/best-practices-in-research-reporting.

Comments from the reviewers:

Reviewer #1: Thanks for the opportunity to review your manuscript. My role is as a statistical reviewer, so my review concentrates on the study design, data, and analysis that are presented. I have put general questions first, followed by queries relevant to a specific section of the manuscript (with a page/line reference).

This manuscript presents a cohort study of people utilising care for Hep C in the country of Georgia from 2015-2020, specifically comparing those also with TB. In this context (Georgia), there has been a program to eliminate Hep C from 2015, with free testing and treatment for citizens. There is evidence that Hep C is more commonly encountered in those with TB, but without formal referral pathways for TB patients. The manuscript describes the cascade of care for Hep C, and investigates personal factors for association with drop-out of the Hep C care system. The study uses data from a notifiable diseases database (TB), a screening registry (Hep C screening), a clinical database (Hep C testing and treatment), and the national death registry. Linkage between the databases was accomplished with deterministic linkage on a national ID. The drop-out was measured as time between a positive test to initiation of treatment, and analysed with a Fine-Grey model. 

Was an analysis plan created for the study? Is this available to see as part of the review? 

I would consider adding the study design (e.g. retrospective cohort study) to the title of the manuscript.

Are there treatment services for TB and Hep C outside of the national programs, e.g. care provided privately by physicians? 

In the methods I would include explicitly how missing data was dealt with in the analysis - e.g. missing as indicator in the LTFU analyses. This makes the assumption that the missing level of a variable is not correlated with any of other of the variables in the analysis (MCAR). Is this a reasonable assumption to make with this data? Is it possible to use an imputation or IPW strategy to be able to make only a missing-at-random assumption? 

P7, L114. Are the TB services offered to everyone, or limited to citizens (like the Hep services)? 

P7, L133. What does 'NTP' stand for?

P10, L191. Was mortality relatively common in the cohort? Is it possible to see the censoring reason (end of follow-up, mortality) by TB status to help assess if competing risks are likely to be an issue. 

P10, L192. To clarify, was a standard cox model used to estimate HRs, rather than a subdistribution HR?

P11, L207. It's good to see covariate selection based on a DAG - is it possible to include this in the supplementary material?

P11, L219. Is this missing ID likely to come from incomplete collection, or because those patients did not have a national ID (e.g. non-citizens)? 

P14, L287. 'multivariable' is the preferred term for this analysis ('multivariate' is used with multiple outcomes or with mixture models) 

P18, L378. While this is a limitation to making a causal inference, I do think there being able to describe gaps between TB and non-TB patients as you have done is still very useful, and that if establishing a treatment gap then inappropriate adjustment of confounders (which effectively end up justifying a gap because someone is older, from a rural location etc.) is usually the problem. 

P29, Table 2. Is Tbilisi the only major urban region in the country? Is there another label for the 'other' category that is more informative? 

P36, Figure 2. What are the error bars that are separate from the bars on these figures? It might be better to include the estimate of number of patients as a dot with error bars placed directly on the dots. I would also consider reporting the cascade as a rate (%) rather than as remaining sample. 

P38, Figure 2. Is it possible to include a confidence interval for the cumulative incidence estimates? 

Reviewer #2: Thankyou for the opportunity to review this paper. The authors have presented an important topic and have well described their results and conclusions. I have put the specific comments in the pdf version of the article. Specifically I would mention that I think the cascade should be in the main body of the article whereas the tables of regression do not need to be ( and can be in the supplement).

The title is good but the initial explanation of the objectives does not reflect that; later in the article it is better reflected. Therefore I would ask the authors to check the text where they have mentioned what the objective of the article is and make sure that it corresponds to the title ( and the later described objectives.

Thanks.

Reviewer #3: Thank you for this well-written and important article. I recommend acceptance with minor revisions.

Methods

- line 191. You mention death was treated as a competing risk. Did you employ competing risk regression? If so, please explain which method (e.g. Fine-Gray) you used and whether your Hazard Ratios are in fact subdistribition Hazard Ratios. 

- Risk factors for LTFU in Tb-patients. Could you also assess these factors in Non-TB patients and compare?

- line 207: you mention directed acyclic graph theory for a-priori covariate selection. Could you provide the graph (e.g. in supplementary material) so readers understand the thought process?

Results

- line 227: Do you have more information (other than age and gender) for those withouth TB co-infection?

- line 244: how many died between testing for viremia and treatment initiaition

- line 250 - : Could you provide statistical tests of significane between the respective proportions of those with and withouth TB? 

- line 264 - : Could you provide statistical tests of significane between the respective median follow-up times between those with and withouth TB?

- Table 2 and 3: Why are no adjusted RR provided for gender?

Discussion

- Could you contrast with data from HCV cascaded of care (non-TB) from other countries?

- Could you discuss the role of HIV and also Injecting drug use in the management of TB and HCV, how this could impact the cascade of care and how integration of care for HIV and injecting drug use with TB and HCV services should ideally by designed and how this is currently done in Georgia.

Reviewer #4: Dear authors, 

Your paper is presenting important set of data which can help shape up policies to address the gaps. I have the following queries:

1) Prisoners are highly vulnerable groups for TB and Hep C. Please elaborate in the text whether the data from TB and Hep C Elimination programmes include individuals who are currently in prisons (the penitentiary sector), (not with the history of incarceration). If so which proportions were prisoners during the dx and tx initiation. 

2) Higher proportions of LTFU and delay in start of treatment after viremia testing among people with TB are important findings. What could be the cofounding factors (e.g. socioeconomic determinants, homeless, alcohol/substance use) and how they are ruled out. Any statistical analysis to that end? If not this could be another limitation of the study. To your view what is the explanation of difference in male and female? how about MSMs?

3) Further clarify in the text which percentage of individuals on the data base had DR-TB

4) Which percentage of triple infection Hep C, HIV and TB exists in your setting? if this data is not available please mention it.

5) The below paper in Lancet Infectious Diseases had highlighted the need for integrated services. The same paper mentions Georgia as a good practice. Which elements of the mentioned practice has been working well and which are those need improvements or changes.

Dara, M., Ehsani, S., Mozalevskis, A., Vovc, E., Simões, D., Avellon Calvo, A., Casabona I Barbarà, J., Chokoshvili, O., Felker, I., Hoffner, S., Kalmambetova, G., Noroc, E., Shubladze, N., Skrahina, A., Tahirli, R., Tsertsvadze, T., & Drobniewski, F. (2020). Tuberculosis, HIV, and viral hepatitis diagnostics in eastern Europe and central Asia: high time for integrated and people-centred services. The Lancet. Infectious diseases, 20(2), e47-e53. https://doi.org/10.1016/S1473-3099(19)30524-9

[LINK]

---

## [Decision Letter · Decision Letter 2]

3 Mar 2023

Dear Dr. Baliashvili,

Thank you very much for re-submitting your manuscript "Hepatitis C care cascade among patients with and without tuberculosis: Nationwide retrospective cohort study in the country of Georgia, 2015-2020" (PMEDICINE-D-22-03278R2) for review by PLOS Medicine.

I have discussed the paper with my colleagues and the academic editor and it was also seen again by xxx reviewers. I am pleased to say that provided the remaining editorial and production issues are dealt with we are planning to accept the paper for publication in the journal.

[LINK]

We look forward to receiving the revised manuscript by Mar 10 2023 11:59PM.   

Sincerely,

Philippa Dodd, MBBS MRCP PhD

PLOS Medicine

plosmedicine.org

Requests from Editors:

GENERAL

Thank you for your detailed and considerate responses to previous editor and reviewer comments. Please see below for further revisions that we require you address in full.

TITLE

Thank you for revising your title suggest “Hepatitis C care cascade among patients with and without tuberculosis between 2015 and 2020: A nationwide observational cohort study from the country of Georgia”

ABSTRACT

Response 13: “…NNT and NNH measures are not relevant in the care cascade analysis.” We agree with you. This information was transposed from a checklist incorrectly. Please accept our apologies.

Line 50: please add “(LTFU)” to define the abbreviation used latterly.

Line 53: “…adults…” please clearly state the total number of participants included in your study here

Line 60-66: “…20%...” of how many? Please clearly define numerators and denominators used to derive percentages to improve clarity for the reader.

Line 71: please define “MDR” here at first use

Line 74: suggest “…we were unable to account for the impact of all confounding factors, such as…” or something similar

AUTHOR SUMMARY

Thank you for including an author summary. Please structure your summary according to PLOS Medicine’s style. Please see our author guidance here https://journals.plos.org/plosmedicine/s/revising-your-manuscript#loc-author-summary

We encourage you to see our website here https://journals.plos.org/plosmedicine/ for examples in published research articles.

The author summary should include ideally 2-3 (but no more than 4) single sentence bullet points for each of the following questions. Bullet points should be objective, brief, succinct, specific, accurate, and avoid technical language.

Why Was This Study Done? 

Authors should reflect on what was known about the topic before the research was published and why the research was needed.

What Did the Researchers Do and Find? 

Authors should briefly describe the study design that was used and the study’s major findings. Do include the headline numbers from the study, such as the sample size and key findings. 

What Do These Findings Mean? 

Authors should reflect on the new knowledge generated by the research and the implications for practice, research, policy, or public health. Authors should also consider how the interpretation of the study’s findings may be affected by the study limitations.

TABLES

Thank you for adding the tables detailing the baseline characteristics of the study population. Please move these to the main manuscript.

SUPPLEMENTARY MATERIAL FOR REVIEWERS

Thank you for including this information. To help facilitate the transparency of data reporting, please include these files as supporting information to be published with the manuscript. Please ensure that files are affiliated to an appropriate caption that clearly describes their contents, including abbreviations. Please report p as <0.001 and where higher as p=0.002, figure SR3 for example

FIGURES

To make your figures, including those in the supporting files, more accessible to those with colour blindness please consider avoiding the use of red and/or green.

DISCUSSION

Line 387: suggest “…with TB between 2015 and 2019...”

REFERENCES

PLOS Medicine does request that for in-text reference callouts there are no spaces between citations. Please amend (publication without is a formatting error, perhaps originating in the citation manager, which we will look into). 

SOCIAL MEDIA

To help us extend the reach of your research, please provide any Twitter handle(s) that would be appropriate to tag, including your own, your coauthors’, your institution, funder, or lab. Please detail any handles you wish to be included when we tweet this paper, in the manuscript submission form when you re-submit the manuscript.

COMMENTS FROM THE ACADEMIC EDITOR

I agree with decision for minor revision.

My only other comment is to have clarity on changes in downstream processes over time. They describe increases in patients completed Ab testing and decreases in proportion who tested positive. Of those who tested positive, did f/u viral load testing change over time? Of those with confirmed HCV infection, did treatment linkage and initiation change over time?

Comments from Reviewers:

Reviewer #1: Thanks for the revised manuscript and responses to my review. The revision covers most of the original queries from the first review.

I agree with the authors that NNH/NNT would not be appropriate applied to a cascade of care study. Also, I support the comment about adjustment for the biological sex variable (i.e. should not be adjusted for). I think the DAGs are a good inclusion and explain the choice of regression models. 

It has been common in the past to rely on a specific threshold (e.g. 5%) for using complete-case analysis or missing as indicator. The issue with this is that the proportion of missing data does not fully capture how much information is lost, it's possible to lose a fairly small amount of missing information that is critical to getting an unbiased effect estimate (https://doi.org/10.1016/j.jclinepi.2019.02.016). For this analysis, there are several 'principled approaches' (https://doi.org/10.1093/aje/kwx348) that could be suitable. For predictors of LTFU in Hep C, and LTFU in HSV care, I would suggest multiple imputation would be reasonably easy to implement (with an MI model using the same covariates as the substantive analysis), with different types of variables (continuous and categorical) MICE would be a good choice. Most software packages now have the capability (e.g. proc MI in SAS, MICE package in R, mi procedures in Stata) and are straightforward to use.

P12, L257. What software was used in the analyses? 

Reviewer #2: Thankyou for the responses and corrections to the first review. I have no further comments on this version. 

Reviewer #3: Thank you for the very thorough revision of the article. I recommend acceptance. 

Reviewer #4: Thanks for your response to queries and having addressed suggestions.

[LINK]

---

## [Decision Letter · Decision Letter 3]

13 Apr 2023

Dear Dr Baliashvili, 

On behalf of my colleagues and the Academic Editor, Professor Amit Singal, I am pleased to inform you that we have agreed to publish your manuscript "Hepatitis C care cascade among patients with and without tuberculosis: Nationwide retrospective cohort study in the country of Georgia, 2015-2020" (PMEDICINE-D-22-03278R3) in PLOS Medicine.

Please address the below prior to publication:

1) Title – please replace “retrospective” with “observational”

2) Abstract line 68 – please define “HR” and “95% CI” at first use 

3) Abstract line 73 – please define “aRR” at first use 

4) Author Summary line 92 – please replace “retrospective” with “observational” and throughout where relevant.

5) Methods line 269 (from the statistical reviewer) – please add the number of imputations that were used.

PRESS

Best wishes,

Pippa 

Philippa Dodd, MBBS MRCP PhD  

PLOS Medicine